# Effects of Light and Oxygen on Chlorophyll *d* Biosynthesis in a Marine Cyanobacterium *Acaryochloris* *marina*

**DOI:** 10.3390/plants11070915

**Published:** 2022-03-29

**Authors:** Yuki Tsuzuki, Yusuke Tsukatani, Hisanori Yamakawa, Shigeru Itoh, Yuichi Fujita, Haruki Yamamoto

**Affiliations:** 1Graduate School of Bioagricultural Sciences, Nagoya University, Nagoya 464-8601, Japan; duffko0614@gmail.com (Y.T.); yamakawa008@gmail.com (H.Y.); fujita@agr.nagoya-u.ac.jp (Y.F.); 2Japan Agency for Marine-Earth Science and Technology (JAMSTEC), Yokosuka 237-0061, Japan; tsukatani@jamstec.go.jp; 3Graduate School of Science, Nagoya University, Nagoya 464-8601, Japan; itoh@bio.phys.nagoya-u.ac.jp

**Keywords:** cyanobacteria, chlorophyll biosynthesis, chlorophyll *d*

## Abstract

A marine cyanobacterium *Acaryochloris* *marina* synthesizes chlorophyll (Chl) *d* as a major Chl. Chl *d* has a formyl group at its C3 position instead of a vinyl group in Chl *a.* This modification allows Chl *d* to absorb far-red light addition to visible light, yet the enzyme catalyzing the formation of the C3-formyl group has not been identified. In this study, we focused on light and oxygen, the most important external factors in Chl biosynthesis, to investigate their effects on Chl *d* biosynthesis in *A. marina*. The amount of Chl *d* in heterotrophic dark-grown cells was comparable to that in light-grown cells, indicating that *A. marina* has a light-independent pathway for Chl *d* biosynthesis. Under anoxic conditions, the amount of Chl *d* increased with growth in light conditions; however, no growth was observed in dark conditions, indicating that *A. marina* synthesizes Chl *d* normally even under such “micro-oxic” conditions caused by endogenous oxygen production. Although the oxygen requirement for Chl *d* biosynthesis could not be confirmed, interestingly, accumulation of pheophorbide *d* was observed in anoxic and dark conditions, suggesting that Chl *d* degradation is induced by anaerobicity and darkness.

## 1. Introduction

Photosynthetic organisms on earth use tetrapyrrole molecules, called chlorophylls (Chls), to absorb light and convert it into chemical energy. It has been reported that there are several types of Chls with different side chains, such as Chl *a*, *b*, *c*, *d*, and *f*, and each has different absorption properties [1]. Chl *a* is the major photosynthetic pigment in almost all oxygen-evolving photosynthetic organisms. In higher plants, Chl *a* and Chl *b* are mainly biosynthesized and the reaction centers of the photosystem contain only Chl *a*, while the antenna complex retains both Chl *a* and Chl *b*. Chl *a* and *b* can be interconverted by a reaction mechanism called the Chl cycle, and the ratio of Chl *a*/*b* is strictly controlled [2]. Chl *c* and its analogs have been reported as an accessory pigment in some secondary symbiotic algae with a porphyrin ring as a backbone [3]. Chl *d* and *f* occur in some cyanobacteria, and these can absorb the longest wavelength of light, close to near-infrared light, among the Chl species, contributing to expanding the niche of light wavelengths available for photosynthesis [4,5]. Chl *f* is known as an auxiliary Chl synthesized by cyanobacteria, such as *Chlorogloeopsis fritschii* PCC 9212 in response to far-red light irradiation [6]. On the other hand, the marine cyanobacterium *Acaryochloris* retains Chl *d* as its major pigment, and is an unusual cyanobacterium that uses Chl *d* other than Chl *a* as its major Chl [4]. Although *Acaryochloris* also retains Chl *a*, more than 90% of the total Chl is composed of Chl *d*, and it has been reported that Chl *d* is coordinated in the photosynthetic reaction centers [7,8].

All Chls, except Chl *c*, are synthesized by a common biosynthetic pathway up to Chl *a* (or chlorophyllide *a*), and then the side chain modification specific to Chl *b*, *d*, and *f* occurs on Chl *a* as a precursor [1,9] (Figure 1). A methyl group at the C7 position of Chl *a* is converted to a formyl group by chlorophyllide *a* oxygenase to synthesize Chl *b*. When the synthesized Chl *b* is reconverted to Chl *a*, the formyl group at the C7 position is converted back to a methyl group through a two-step reaction of Chl *b* reductase and 7-hydroxymethyl Chl *a* reductase [2]. Recently, it has been shown that the formation of a formyl group at the C2 position in Chl *f*, which determines its absorption properties, is catalyzed by PsbA4 (ChlF), a paralogue of the PSII core subunit, D1 protein [10]. This reaction has been reported to proceed in a light-dependent manner; however, the detailed reaction mechanism is not well understood [11]. While the enzyme responsible for the formation of the formyl group at the C3 position in Chl *d* has not been identified, its formation is expected to be a multi-step catalytic reaction involving the removal of a carbon atom (C3^2^) from a vinyl group at the C3 position and the formation of an oxo group to form a formyl group. A previous report using isotope ^18^O proposed that the oxygen in the formyl group at the C3 position of Chl *d* is derived from molecular oxygen rather than water molecules suggesting this formylation catalyzed by an oxygenase-type enzyme [12].

In this study, to clarify the effects of light irradiation and environmental oxygen level on Chl *d* biosynthesis in *Acaryochloris*, we investigated dark-heterotrophic and anoxic cultures of *Acaryochloris*, respectively, and determined their respective Chl compositions.

## 2. Results

### 2.1. Heterotrophic Growth with Carbon Sources

To search for organic compounds supporting dark heterotrophic growth of *Acaryochloris* strains, we tested four carbon sources, glycerol, sucrose, glucose, and acetic acid, using six *Acaryochloris* strains, *Acaryochloris marina* MBIC 11017 (Am11017), *Acaryochloris* sp. MBIC 10679 (Asp10679), MBIC 10690 (Asp10690), MBIC 10696 (Asp10696), MBIC 10697 (Asp10697), and MBIC 10699 (Asp10699) (Figure 2). Significant dark heterotrophic growth was observed in five strains except for Asp10690; Am11017 and Asp10696 showed dark heterotrophic growth in the medium supplemented with glycerol, Asp10699 and Asp10679 with sucrose, and Asp10697 with glucose, respectively. Acetic acid did not support dark heterotrophic growth in any strains. Under mixotrophic conditions, in which the addition of carbon sources and light irradiation are simultaneous, growth inhibition was found for several combinations of carbon sources and *Acaryochloris* strains. Severe growth inhibition was observed in all strains except Am11017 by the addition of glycerol, in Am11017, Asp10690, and Asp10697 by the addition of sucrose, and in Am11017 and Asp10690 by the addition of glucose. In the other conditions, the addition of each carbon source promoted the growth.

### 2.2. Biosynthesis of Chl in Dark Heterotrophic Growth

To investigate the light requirement of Chl *d* biosynthesis, Am11017 was heterotrophically grown in the dark using glycerol, and Chl *a* and Chl *d* amounts were determined (Figure 3). Although the growth rate was slightly slower in the dark heterotrophic conditions than in the photoautotrophic condition, the cell density increased approximately eight-fold in 16 days even in the dark heterotrophic condition (Figure 3A). Under the photoautotrophic condition, Chl *a* and Chl *d* per cell increased with growth while maintaining the Chl *a*/*d* ratio at about 0.05 (Figure 3B,C). In the dark heterotrophic conditions, both Chl *a* and Chl *d* also increased, but the increase in Chl *d* was less, and, conversely, the increase in Chl *a* was higher than those in the photoautotrophic condition. As a result, the Chl *a*/*d* ratio increased to 0.39 in the dark heterotrophic conditions compared to 0.05 in the photoautotrophic condition (Figure 3C). In other words, the amount of Chl *a* increased to about 28% of the total Chls in 16 days under the dark heterotrophic condition. Although Chl *a*/*d* ratio increased, the amount of cellular Chl *d* increased significantly under dark heterotrophic conditions, indicating that Chl *d* biosynthesis can proceed in a light-independent manner in Am11017.

### 2.3. Biosynthesis of Chl d under Anoxic Conditions

To examine the biosynthesis of Chl *d* under anoxic conditions in Am11017, the anoxic cultivation was performed using the oxygen-absorbing pouch in a sealed container. This pouch absorbs oxygen and releases CO_2_ simultaneously, and it was found that the pH of IMK medium is decreased by high CO_2_ concentration (data not shown). Considering the possibility that the decrease in pH affects the growth of *Acaryochloris*, we attempted to stabilize the pH of the IMK medium by adding buffers. By fixing the pH of IMK medium in the range of 6–12 with a buffer, the optimum pH for the growth of Am11017 was verified, and it was found that the growth rate was highest at pH 9.0 (Appendix A). Although the optimum pH was 9.0, we adjusted the pH to 8.5, which is the original pH of IMK medium, and evaluated the growth rate of Am11017 and the amounts of Chl *d* and Chl *a* under anoxic light and dark conditions, respectively. In the anoxic and light condition, the Am11017 cells multiplied more than four-fold in 7 days, and the amount of Chl *d* and Chl *a* also increased with the Chl *a*/*d* ratio maintained at about 0.08 (Figure 4). In contrast, in the anoxic and dark condition, OD_750_ decreased to 1/3 in 7 days despite the addition of glycerol, and the amount of Chl *d* and Chl *a* also decreased, while the Chl *a*/*d* ratio remained about 0.08 (Figure 4). Additionally, there was found to be an accumulation of a unique pigment, different from Chl *a*, *d,* and carotenoids, in anoxic dark cultured Am11017 cells. This pigment eluted at 6.6 min in HPLC chromatogram (Figure 5A, orange arrow) and the in-line absorption spectrum of this pigment showed a Qy peak at 693 nm similar to that of Chl *d* with additional absorption peaks at 518 and 551 nm, and Soret bands at 384 and 422 nm (Figure 5B). This Chl *d*-like unknown pigment was detected from the second day of anoxic dark cultivation, and continued to increase over time until the seventh day (Figure 5A). From these results, in the anoxic and light condition, Chl *d* and Chl *a* biosynthesis proceeded in the same way as the oxic condition, while under the anoxic and dark condition, Am11017 did not grow at all and de novo biosynthesis of Chl *d* and Chl *a* was not observed. Additionally, Chl *d*-like unknown pigment was accumulated only under anoxic and dark conditions.

### 2.4. Accumulation of Pheophorbide d

A Chl *d*-like unknown pigment was detected in anoxic and dark conditions. The absorption spectrum of the pigment seems similar to that of Chl *d*, while it has additional absorption peaks and shifted Soret peaks (Figure 5B). The appearance of the peaks at 500–550 nm and the shift in the Soret peak are also observed in pheophytin *a*, in which Mg is desorbed from Chl *a*. In addition, the HPLC condition was reversed-phase column chromatography with elution starting from the least hydrophobic compound. The retention time of the unknown pigment, 6.6 min was close to that of pheophorbide *a* (eluted at 14 min), which has a lower hydrophobicity due to the cleavage of a phytol chain than pheophytin *a* (eluted at 26 min). This relationship between retention time and hydrophobicity suggests that the unknown pigment is pheophorbide *d* (Pheide *d*), which is Mg-depleted and dephytylated from Chl *d*. To identify this unknown pigment, Pheide *d* standard was prepared and applied to HPLC analysis under the same condition. The Pheide *d* standard and the unknown pigment eluted at the same retention time and their in-line absorption spectra were identical (Figure 6A,C). Additionally, the molecular-ion mass numbers of the pigment that eluted at 6.6 min were 592.85 Da as [M − H]^−^ and 628.85 Da as [M + Cl]^−^, which were very close to those of the Pheide *d* standard (Figure 6B,D). These values are considered identical within error to the calculated mass numbers of Pheide *d* (C_34_H_34_N_4_O_6_), 593.24 Da as [M − H]^−^, and 629.22 Da as [M + Cl]^−^, suggesting that they are ascribable to Pheide *d*. These results indicate that Am11017 grown under anoxic and dark conditions accumulates Pheide *d*.

## 3. Discussion

In this study, we performed dark-heterotrophic culture of *Acaryochloris* to evaluate Chl *d* biosynthesis in the dark. A significant amount of Chl *d* was synthesized even in the dark, suggesting that *Acaryochloris* has a light-independent Chl *d* synthetic pathway. However, in the anoxic condition, cell growth and Chl *d* synthesis occurred only with light irradiation, and putative Chl *d* degradate, pheophorbide *d*, accumulated in the absence of light irradiation.

### 3.1. Differences in Carbon Requirement for Heterotrophic Growth

We found that five of the six *Acaryochloris* species grew in the dark heterotrophic conditions when appropriate organic compounds were supplied (Figure 2). Additionally, some strains were inhibited in growth under mixotrophic conditions with the carbon sources. These carbon sources that inhibited mixotrophic growth also did not support dark heterotrophic growth in the same strain (exceptionally, Asp10696 showed slight dark heterotrophic growth with glycerol). This phenomenon of growth inhibition by exogenous carbon sources was also observed in *Synechocystis* sp. PCC6803 for glucose. Through the analysis of the glucose transporter GlcP and the glucokinase (glk), the accumulation of unmetabolized glucose in the cells caused the growth defect [13]. Additionally, it has been reported that the loss of FesM, a protein involved in PSI cyclic electron transfer, in *Synechococcus* sp PCC 7002 prevents mixotrophic growth using glycerol, suggesting that the addition of glycerol affects the reduction state of the PQ pool, leading to poor growth [14]. In this experiment, the growth of *Acaryochloris* species was inhibited by the addition of glycerol, sucrose, and glucose, suggesting that the influx of reducing power from these exogenous carbon sources affects the reduction state of the PQ pool, which resulted in inhibited growth (Figure 2). In the other mixotrophic conditions where growth was not inhibited, growth was accelerated compared to the condition without a carbon source, suggesting that these carbon sources were metabolized and used for growth in *Acaryochloris* (Figure 2). Therefore, under these mixotrophic conditions, the balance between catabolism of carbon sources and electron transfer by photosynthesis is maintained. Interestingly, the relationship between the availability of glycerol and glucose for mixotrophic growth was exclusive, with Am11017 tolerating glycerol but not glucose, and vice versa for Asp10679, Asp10696, Asp10697, and Asp10699. This classification correlates with the presence or absence of the photosynthetic antenna complex, phycobilisome, and it will be interesting to see how tolerance to these carbon sources relates to differences in photosynthetic antennas.

Looking at the dark heterotrophic conditions, the carbon sources available to each *Acaryochloris* strain were very different. Acetic acid was available for mixotrophic growth in all strains, but could not support dark heterotrophic growth in all strains. Although acetic acid is expected to provide acetyl CoA for the TCA cycle [15], it is suggested that this alone is not be sufficient to support dark heterotrophic growth in *Acaryochloris*. Since the pentose phosphate pathway is considered as the main carbon metabolism pathway in cyanobacteria, it is possible that the supply of acetyl CoA, which is not a direct substrate for this pathway, did not provide sufficient reducing power. On the other hand, glucose, sucrose, and glycerol provide substrates for the pentose phosphate pathway via the Calvin cycle in their metabolism, and thus can be considered to have provided sufficient reducing power even under complete darkness.

### 3.2. Light and Oxygen Requirement on Chl d Biosynthesis

The amount of Chl *d* in dark-grown Am11017 was examined, and we found that de novo Chl *d* biosynthesis proceeded even in the complete dark condition (Figure 3B). The light requirement of Chl *d* synthesis should be discussed in two reactions, formyl-group modification at the C3 position, and protochlorophyllide (Pchlide) reduction. Chl *f*, which can absorb far-red light similar to Chl *d*, is a modified a methyl-group at the C2 position to a formyl-group in a light-dependent manner by the PsbA paralogue ChlF [10]. In this experiment, Am11017 biosynthesized Chl *d* even in complete darkness, indicating a light-independent synthetic pathway for Chl *d* biosynthesis in *Acaryochloris*. It is thought that the formylation at the C3 position is catalyzed by a completely different mechanism from that of ChlF. Another light-dependent reaction is the Pchlide reduction, which is the rate-limiting reaction in the late stage of the Chl biosynthetic pathway. Two distinct enzymes with different evolutionary origins exist for this reaction: one is known as light-dependent Pchlide reductase (LPOR) and the other is light-independent Pchlide reductase (DPOR) [16]. In angiosperms and some eukaryotic algae, which retain only LPOR, the Pchlide reduction is completely light-dependent and does not proceed in the dark, resulting in the accumulation of Pchlide [17,18]. In cyanobacteria, including *Acaryochloris*, both LPOR and DPOR are conserved, and DPOR promotes the reduction of Pchlide in the dark. In this experiment, both Chl *a* and Chl *d* were synthesized normally and Pchlide was not accumulated in the dark-grown cultures (Figure 3B), indicating that light-independent DPOR is fully functional in *Acaryochloris*.

To examine the effect of oxygen concentration on Chl *d* biosynthesis, *Acaryochloris* was grown in anoxic conditions. The results showed that anoxic cultures of *A. marina* grew normally under light irradiation, but not at all under dark conditions (Figure 4A). In the anoxic and dark condition, heterotrophic growth is expected to be difficult due to the lack of oxygen, the final electron acceptor in the respiratory electron transfer system, caused by low oxygen concentration. Therefore, it is difficult to evaluate the biosynthesis of Chl *d* under such extremely low oxygen conditions because the growth of *A. marina* is inhibited under that condition. However, under anoxic and light conditions, it is possible to produce oxygen from PSII by light irradiation; therefore, this condition should be “micro-oxic” rather than anoxic, and can be used to evaluate Chl *d* biosynthesis under micro-oxic conditions. In addition to Chl *d* biosynthetic enzymes, it has been reported that there are several oxygen-dependent enzymes in the cyanobacterial Chl biosynthetic pathway and how these reactions are affected in a micro-oxic environment. Oxygen-dependent HemF and oxygen-independent HemN have been reported for coproporphyrinogen III oxidation [19,20]. In *Synechocystis* sp. PCC6803 (here after *Synechocystis*), when the respective deficient mutants of *hemF*/*N* were grown under oxic and micro-oxic conditions, *∆hemF* strain could not grow under oxic conditions, while *∆hemN* grew under both conditions but accumulated significant coproporphyrinogen III under the micro-oxic conditions [21]. Since both enzymes, HemF (AM1_0615) and HemN (AM1_0467), are conserved in *A. marina*, HemN can catalyze this reaction in an oxygen-independent manner under micro-oxic conditions in *A. marina*. Another oxygen-dependent reaction is Mg-protoporphyrin monomethyl-ester oxidation (cyclization) and two enzyme systems, oxygen-dependent AcsF and oxygen-independent BchE (ChlE), have been reported for this reaction [22,23]. With a few exceptions, ChlE is not conserved in most cyanobacteria including *Acaryochloris*, and it is thought that this cyclization is catalyzed solely by oxygen-dependent AcsF type enzyme in these cyanobacteria [24]. It has been reported that two homologs of AcsFs (ChlAI and ChlAII) are involved in this reaction in *Synechocystis*, and although ChlAI is the main enzyme, ChlAII is induced under micro-oxic conditions to support the reaction under low oxygen conditions [25]. Two AcsF homologs, AM1_2295 and AM1_0465, are also conserved in *A. marina* and show 77% and 63% homology with ChlAI and ChlAII of *Synechocystis*, respectively. Thus, it is likely that ChlAII supports this reaction under micro-oxic conditions in *Acaryochloris* as well as in *Synechocystis*. This oxygen-dependent reaction seems to proceed normally even under micro-oxic conditions, suggesting that the amount of oxygen supplied from PSII by light irradiation is sufficient for this cyclization to proceed. For the formylation reaction at the C3 position of Chl *d*, previous experiments using oxygen isotope ^18^O suggest that this reaction is catalyzed by oxygen-dependent oxygenase [12]. Since Chl *d* biosynthesis under completely anoxic conditions was not evaluated in this experiment, it is not known whether oxygen-independent enzymes for Chl *d* biosynthesis exists. However, Chl *d* biosynthesis was carried out normally even under micro-oxic conditions as well as under oxic conditions, and no accumulation of precursors was observed. It suggests the existence of a mechanism by which Chl *d* biosynthesis does not stagnate even with a limited oxygen supply provided from PSII, as in the case of HemF/N and AcsF/ChlE described above. Whether there is an oxygen-independent Chl *d* biosynthetic enzyme or whether Chl *d* biosynthesis is carried out by using multiple homologous enzymes corresponding to the oxygen concentration remains to be verified.

### 3.3. Accumulation of Pheide d and Degradation of Chl d

In the anoxic and dark conditions, significant accumulation of Pheide *d* was observed, in which Mg and a phytol chain were detached from Chl *d* (Figure 5 and Figure 6). Since cell growth did not occur in this condition and a decrease in the amount of cellular Chl *d* was observed, this Pheide *d* is considered to be derived from the degradation of Chl *d*. Chl *a* degradation pathway in higher plants is initiated by Mg desorption, followed by phytol chain cleavage, to produce Pheide *a*, and further degradation reactions proceed [26]. At this time, Chl *b* is always reconverted to Chl *a* by converting the formyl group at the C7 position back to a methyl group, and then proceeds through the same degradation pathway as Chl *a* (Chl cycle) (Figure 1) [2]. Studies on the degradation of Chl in cyanobacteria are limited and have not progressed much. However, enzymes which convert Chl *b* to Chl *a* have been recently identified in the Chl *b*-containing cyanobacteria *Prochlorothrix hollandica* and *Acaryochloris thomasi* RCC1774, suggesting that a Chl degradation pathway exists in cyanobacteria the same as in higher plants [27]. In this study, the accumulation of Pheide *d* was observed in *Acaryochloris* cells. While *Acaryochloris* does not conserve the Mg-dechelatase known in higher plants, and it is unclear what mechanism is responsible for Mg desorption, it retains enzymes homologous to Chl dephytylase, which cleave phytol chain from Chls [28]. If Pheide *d* is an intermediate in the Chl *d* degradation pathway, it suggests that the degradation of Chl *d* does not undergo the conversion of Chl *d* to Chl *a* and maintains the formyl group at the C3 position in a pheophorbide form. In higher plants, Pheide *a* is ring opened by pheophorbide *a* oxygenase (Pao) and then converted to pFCC by red-Chl catabolite reductase [29]. In *A. marina* genome, several genes with significant homology to the *pao* gene have been found, and it is possible that these enzymes catalyze the ring-opening reaction of Pheide *d*. Accumulation of pheophorbides is an unusual example in cyanobacteria, *A. marina* provides clues to the cyanobacterial Chl degradation pathway.

### 3.4. Increase in Chl a Content in the Dark-Heterotrophic Condition

The amount of Chl *a* in *A. marina* increased under the dark heterotrophic conditions; compared with the photoautotrophic conditions, the Chl *a*/*d* ratio increased about eight-fold from 0.05 to 0.39, and Chl *a* accumulated up to 28% of the total Chl in the cells (Figure 3B,C). The Chl *a*/*d* ratio of the Asp10699 strain was the same in the dark heterotrophic conditions as in the photoautotrophic condition (Appendix A). These results suggest that the increase in Chl *a* amount in the dark heterotrophic condition is not a general phenomenon in *Acaryochloris*, but is unique to *A. marina* MBIC11017. In cyanobacteria, Chl is mainly bound to the photosynthetic reaction center complexes PSI and PSII. However, in *Acaryochloris*, in addition to PSI and PSII, Pcb (Prochlorophyte Chl-binding protein), a Chl-binding antenna complex, has been reported as a third Chl-binding protein complex [30,31]. Pcb binds both Chl *a* and Chl *d*, and the ratio of Chl *a*/*d* in Pcb varies depending on the culture conditions of *A. marina* [30]. It was also reported that Pcb from *Acaryochloris* bound Chl *a* when expressed in the other cyanobacterium *Synechocystis*, which contains only Chl *a* [32]. It is possible that the increase in the amount of Chl *a* under the dark heterotrophic conditions was caused by a change in the Chl *a*/*d* ratio in the Pcb complex in *A. marina*. Another possibility is that Chl *a* binds to a site where Chl *d* was originally coordinated in PSI and PSII, resulting in a higher Chl *a*/*d* ratio. In higher plants, photosystem reaction centers coordinate only Chl *a*, but in strains with abnormal accumulation of Chl *b* due to heterologous expression of chlorophyllide *a* oxygenase from *Proclothrix*, it has been reported that Chl *b* is also coordinated in PSI complexes [33]. This result indicates that although Chl-binding proteins such as PSI selectively coordinate Chl *a* and Chl *b*, abnormal changes in the Chl *a*/*b* ratio can cause Chl *b* to replace the site where Chl *a* originally bound. The Chl *a*/*d* ratio in PSI and PSII complexes from *A. marina* was reported to be about 0.014 and 0.03, respectively [34,35]. However, if Chl *a* accumulates in the dark heterotrophic cultures and Chl *a*/*d* ratio changes in *A. marina*, it is possible that Chl *a* binds to the site where Chl *d* normally coordinates in PSI and PSII. To confirm these hypotheses, spectroscopic analysis of dark-grown *A. marina* cells will be necessary.

## 4. Materials and Methods

### 4.1. Acaryochloris Strains and Culture Conditions

*Acaryochloris* species were purchased from NBRC as *Acaryochloris marina* NBRC 102967 (formerly MBIC 11017) [4], *Acaryochloris* sp. NBRC 102863 (MBIC 10679), NBRC 102866 (MBIC 10690), NBRC 102868 (MBIC 10696), NBRC 102869 (MBIC 10697), and NBRC 102871 (MBIC 10699). Contaminant bacteria, which show small white spherical colonies, in the original cultures were eliminated by repeated single-colony isolation and agar plate cultures in laboratory. These strains were grown in a 0.5% (*v*/*v*) Daigo IMK medium with 3.6% (*w*/*v*) artificial seawater (Marine Art SF-1; Osaka Yakken Co., Ltd., Osaka, Japan) with fluorescent light exposure at 10 µE/m^2^ s (FRL40SW; Hitachi Global Life Solutions, Inc., Tokyo, Japan) at 26 °C. For experiments to verify the optimal pH, 20 mM HEPES or 20 mM CAPS was added to the IMK medium to stabilize the culture pH in the range of 6–12. In the mixotrophic and dark-heterotrophic conditions, 30 mM glycerol, 5 mM sucrose, 5 mM glucose, and 15 mM acetic acid were added to the medium, respectively. Anoxic cultivation was carried out in a sealed container (AnaeroPack Rectangular Jar; Mitsubishi Gas Chemical Co., Tokyo, Japan) with oxygen absorber and carbon dioxide generator (AnaeroPouch Kenki 5%; SUGIYAMA-GEN CO., Ltd., Tokyo, Japan).

### 4.2. Pigment Extraction and HPLC Analysis

Total pigment from *Acaryochloris* cells was extracted according to the previous report [16]. After adding methanol to the collected cells, the cells were disrupted by sonication and then left on ice for 30 min for pigment extraction. The supernatant collected by centrifugation at 15,000 rpm for 20 min (MX-300, AR015-24; TOMY, Tokyo, Japan) and used for HPLC analysis. HPLC analysis (LC20-AD; Shimadzu, Kyoto, Japan) was performed according to Zapata’s method [21,36] and Chl *a*, *d*, and pheophorbide *d* were detected by absorption at 440 nm and 690 nm. Quantification of Chl *a* and *d* was performed by preparing standard curves of peak areas on HPLC chromatograms with standard pigments.

### 4.3. Synthesis of Pheophorbide d

The standard of pheophorbide *d* was prepared from Chl *d*. Crude Chl *d* mixture was extracted from *A. marina* cells using acetone. The phytol chain removal and Mg desorption reactions in Chl *d* were in accordance with previous studies [37]. The minor modifications are described below. To cleave phytol chains, chlorophyll dephytylase from *Synechococcus elongatus* PCC7942 (synpcc7942_2532) was used. Heterologous expressed His-tagged dephytylase was purified using Ni^2+^ affinity column (HisTrap^TM^ HP; Cytiva, Tokyo, Japan) from *E. coli*. The crude Chl *d* mixture was incubated with the purified dephytylase at 30 °C for 2 h in the dark. After the phytol cleavage reaction with dephytylase, the reaction mixture was acidified with 1 M HCl to remove the central Mg of chlorophyllide *d* followed by acetone/hexane phase separation. Pheophorbide *d* in the acetone phase was collected in the diethyl ether phase and then dried to give a pheophorbide *d* standard.

### 4.4. LC-MS Measurement

The LC-MS system consisted of a C18 reverse-phase column (Synergi Hydro-RP 4 µm, 4.6 φ × 150 mm; Shimadzu GLC Ltd., Tokyo, Japan), a photodiode-array spectrophotometer detector (SPD-M20A; Shimadzu, Kyoto, Japan) and a LCMS-2010EV quadrupole mass spectrometer equipped with an electrospray ionization (ESI) probe (Shimadzu, Kyoto, Japan). The following method is based on a previous report [38]. The mobile phase was methanol:acetonitrile:monoethanolamine (pH 7.0) = 52.5:30.0:17.5 (*v*/*v*/*v*), and the flow rate was isocratic at 1 mL/min. The ESI-MS settings were as follows: capillary temperature; 230 °C, sheath gas (N_2_) pressure; 0.1 MPa, and spray voltage; 2.0 kV (negative-ion ESI). The column oven was set at 40 °C. Pigment extracts were treated with 10% acetic acid, which increased separation capacities for the pigments on the Synergi reverse-phase column. The treated pigment mixtures were filtered through a 0.45-mm filter before injection.

## Figures and Tables

**Figure 1 plants-11-00915-f001:**
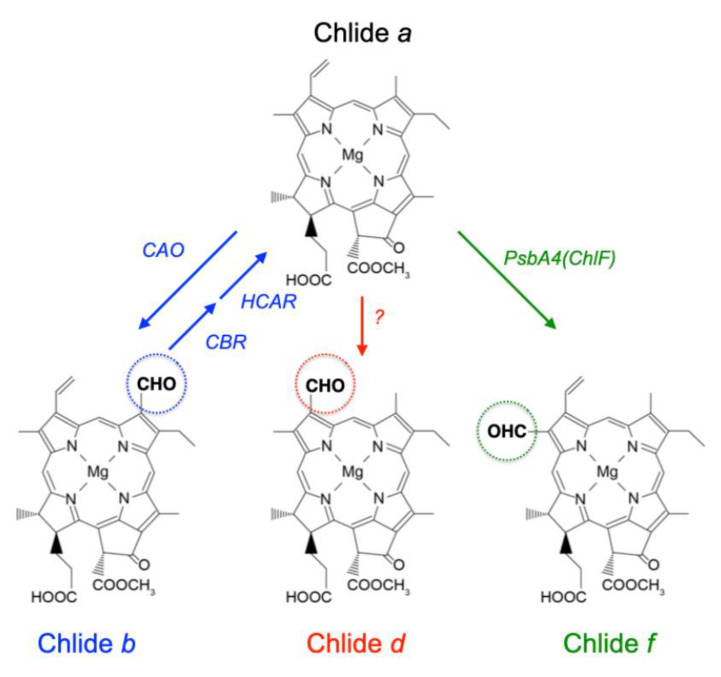
Chl *a* derivatives. Chl metabolic pathways for Chl *b*, *d*, and *f* from Chl *a*. The dashed circle indicates modified position and arrows indicate each of the following enzymatic reactions. CAO; chlorophyllide *a* oxygenase, CBR; Chl *b* reductase, HCAR; 7-hydroxymethyl Chl *a* reductase.

**Figure 2 plants-11-00915-f002:**
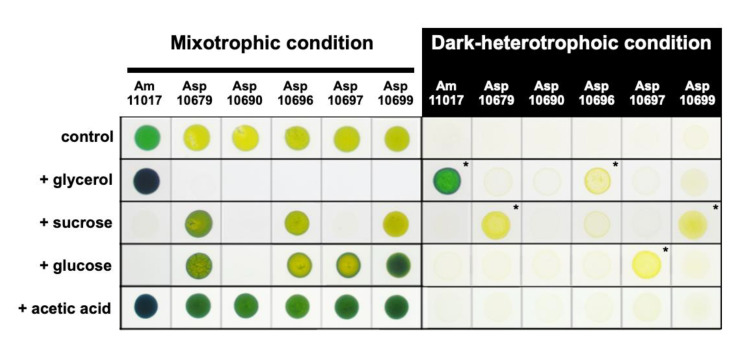
Mixotrophic and Heterotrophic growth of *Acaryochloris* species. Six *Acaryochloris* species (*A. marina* MBIC11017; Am11017, *A*. sp MBIC10679; Asp10679, *A*. sp MBIC10690; Asp10690, *A*. sp MBIC10696; Asp10696, *A*. sp MBIC10697; Asp10697, *A*. sp MBIC10699; Asp10699) grew with several carbon sources under light or dark condition for 20 days. Control means without carbon sources. Asterisks indicate significant growth under the dark-heterotrophic conditions.

**Figure 3 plants-11-00915-f003:**
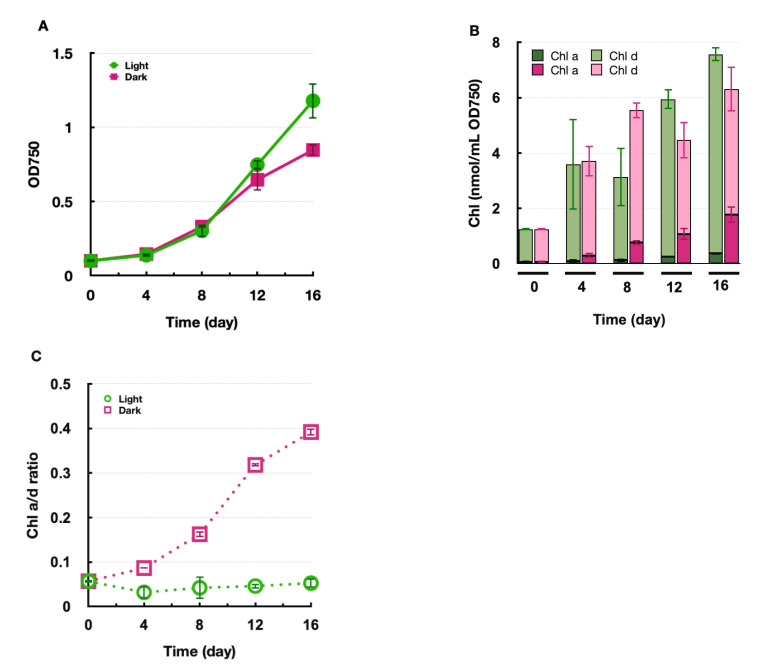
Dark heterotrophic growth and Chl *a*/*d* content in *A. marina.* (**A**) *A. marina* was grown in liquid IMK medium under photo-autotrophic conditions (green circle) and dark-heterotrophic conditions (pink square). (**B**) The amounts of Chl *a* (dark green/dark pink) and Chl *d* (green/pink) were measured on days 0, 4, 8, 12, and 16 of cultivation in photo-autotrophic (green) and dark-heterotrophic (pink) conditions, respectively. (**C**) The ratio of cellular Chl *a*/*d* amount in photo-autotrophic conditions (green circle) and dark-heterotrophic conditions (pink square).

**Figure 4 plants-11-00915-f004:**
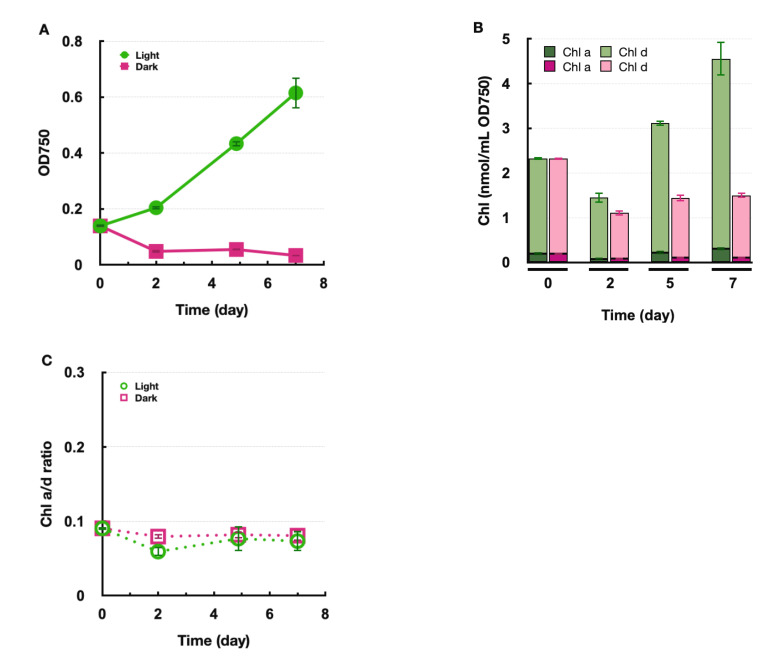
Anoxic growth and Chl *a*/*d* content in *A. marina.* (**A**) *A. marina* was grown under anoxic and light conditions (green circle) and anoxic and dark conditions (pink square). (**B**) The amounts of Chl *a* (dark green/dark pink) and Chl *d* (green/pink) were measured on days 0, 2, 5, and 7 of cultivation in anoxic light (green) and anoxic dark (pink) conditions, respectively. (**C**) The ratio of cellular Chl *a*/*d* amount in anoxic light conditions (green circle) and anoxic dark conditions (pink square).

**Figure 5 plants-11-00915-f005:**
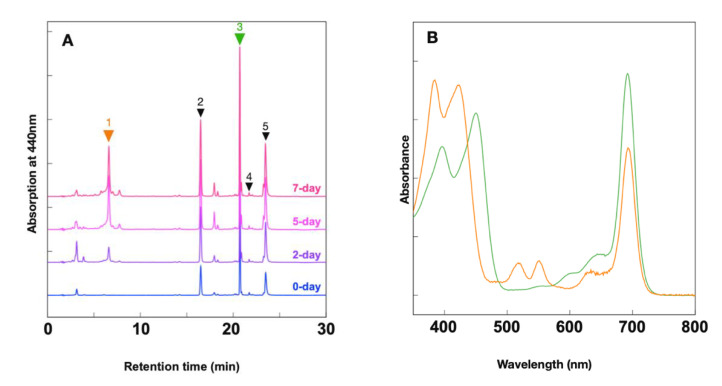
Accumulation of Chl *d*-like pigment in anoxic and dark conditions. (**A**) Total pigment was extracted from *A. marina* cells grown under anoxic and dark conditions at 0 (blue), 2 (purple), 5 (pale violet), 7-day (pink), respectively. The total pigments were separated by C8 column monitored by absorption at 440 nm. Each arrows indicate Chl *d*-like pigment (1), zeaxanthin (2), Chl *d* (3), Chl *a* (4), and α-carotene (5), respectively. (**B**) In-line absorption spectra of Chl *d* (green) and Chl *d*-like pigment (orange).

**Figure 6 plants-11-00915-f006:**
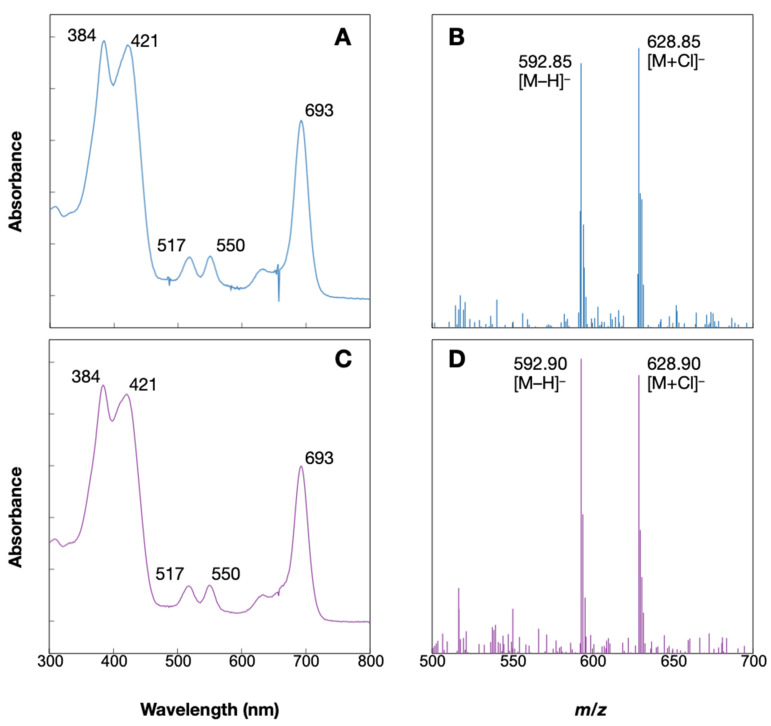
Identification of accumulated Chl *d*-like pigment. In-line absorption spectra (**A**) and mass spectra (**B**) of the 6.6 min elution peak in Figure 5A are shown. Panels (**C**,**D**) represent in-line absorption and mass spectra of the Pheide *d* standard, respectively.

## Data Availability

Data is contained within this article and the Appendix A.

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
