# Peer review of "Effects of Light and Oxygen on Chlorophyll d Biosynthesis in a Marine Cyanobacterium Acaryochloris marina"

_plants, 2022, doi:10.3390/plants11070915_

Round 1
Reviewer 1 Report
The authors reported the effects of light and oxygen on pigment compositions in a cyanobacterium Acaryochloris marina. The results in this paper will help elucidation of the biosynthesis of chlorophyll (Chl) d, and thus this paper will be acceptable after minor revision.
The reviewer has some comments as follows.
1) Figure 3, Figure 4, and Figure S1. Please show the time-dependent changes of Chl a/d ratio.
2) Figure 5. Please assign major fractions in the HPLC chromatograms, such as the fractions at about 17 min and 23 min.
3) Figure 6. Please show the calculated values of [M+H] and [M+Cl] of Chl-d like pigment. Chlorine has two major isotope, 35Cl and 37Cl. Which isotope contains in the Cl-adduct of Chl-d like pigment? The m/z values of both [M+H] and [M+Cl] adducts seems to be different from the calculated values.
4) Lines 287-289. The authors seem to propose oxygen-independent Chl d biosynthesis. However, the usage of oxygen from PS II is possible. Please discuss such possibility of the oxygen-dependent biosynthesis of Chl d or describe the reason for no possibility of the oxygen-dependent biosynthesis of Chl d.
5) Section 3.3. Does Acaryochloris have genes that are homologous with those of Mg-dechelatase and pheophytinase?
Minor comments
The reviewer thinks that there are some redundant descriptions in this paper. For example, in Introduction, the description about Chl f in the second paragraph seems to overlap with that in the first paragraph. The description about Chl-d like pigment in the section 2.3 also seems to overlap with that in the section 2.4. Re-check of the whole text will improve this paper and will be helpful for readers.
Author Response
1) Figure 3, Figure 4, and Figure S1. Please show the time-dependent changes of Chl a/d ratio.
We added additional graphs showing the time-dependent changes of Chl a/d ratio in Figure 3, 4, and S1.
2) Figure 5. Please assign major fractions in the HPLC chromatograms, such as the fractions at about 17 min and 23 min.
We assigned five major peaks in the HPLC chromatogram in Fig. 5.
3) Figure 6. Please show the calculated values of [M+H] and [M+Cl] of Chl-d like pigment. Chlorine has two major isotope, 35Cl and 37Cl. Which isotope contains in the Cl-adduct of Chl-d like pigment? The m/z values of both [M+H] and [M+Cl] adducts seems to be different from the calculated values.
The calculated mass numbers of pheophorbide d (C34H34N4O6; exact mass is 594.25 Da) are 593.24 Da as [M-H]- and 629.22 Da as [M+Cl]-. The actual experimental results showed 592.85 Da and 628.85 Da, respectively, and these values are considered identical within error to the calculated values. From subtracting the value of [M-H]- from value of [M+Cl]-, the isotope of Cl is considered to be 35Cl. We added a sentense to indicate the calculated mass numbers of pheophorbide d.
4) Lines 287-289. The authors seem to propose oxygen-independent Chl d biosynthesis. However, the usage of oxygen from PS II is possible. Please discuss such possibility of the oxygen-dependent biosynthesis of Chl d or describe the reason for no possibility of the oxygen-dependent biosynthesis of Chl d.
As you mentioned, oxygen is availabe from PSII in micro-oxic condition. We pointed out that there may be a mechanism by which Chl d biosynthesis does not stagnate even with a limited oxygen supply provided from PSII. We revised sentences as follows.
Since Chl d biosynthesis under completely anoxic conditions was not evaluated in this experiment, it is not known whether oxygen-independent enzymes for Chl d biosynthesis exists. However, Chl d biosynthesis was carried out normally even under micro-oxic condition as well as under oxic conditions, and no accumulation of precursors was observed. It suggests the existence of a mechanism by which Chl d biosynthesis does not stagnate even with a limited oxygen supply provided from PSII, as in the case of HemF/N and AcsF/ChlE described above.
5) Section 3.3. Does Acaryochloris have genes that are homologous with those of Mg-dechelatase and pheophytinase?
We added sentences describing the presence or absence of Mg-dechelatase and pheophytinase in A. marina into the section. There is no homologue of Mg-dechelatase in A. marina, but there are several proteins that show significant homology to the chlorophyll dephytylase of Camellia sinensis.
Minor comments
The reviewer thinks that there are some redundant descriptions in this paper. For example, in Introduction, the description about Chl f in the second paragraph seems to overlap with that in the first paragraph. The description about Chl-d like pigment in the section 2.3 also seems to overlap with that in the section 2.4. Re-check of the whole text will improve this paper and will be helpful for readers.
Thank you for your valuable advices regarding the description of this paper. We have checked the entire text, focusing on the two paragraphs you pointed out, and revised to eliminate duplication of information.
Reviewer 2 Report
The present article "Effects of light and oxygen on chlorophyll d biosynthesis in a marine cyanobacterium Acaryochloris marina" is quite interesting and well designed. The overall presentation of the article is nice. So can be considered for publication in this journal.
Author Response
The present article "Effects of light and oxygen on chlorophyll d biosynthesis in a marine cyanobacterium Acaryochloris marina" is quite interesting and well designed. The overall presentation of the article is nice. So can be considered for publication in this journal.
Thank you for reviewing our paper.
Reviewer 3 Report
The paper describes chlorophyll (Chl) d synthesis in marine cyanobacterium Acaryochloris marina. Chl d absorbs far-red light in addition to visible light. The effects of light and oxygen were studied. The paper reads well and the figures are clear and informative.
It is interesting that the cyanobacteria produce Chl d in the dark, only to then degrade it into Pheide d – limiting production of Chl d would be more efficient. Authors might like to comment.
Author Response
It is interesting that the cyanobacteria produce Chl d in the dark, only to then degrade it into Pheide d – limiting production of Chl d would be more efficient. Authors might like to comment.
Thank you for your comment. We also think it is a very interesting phenomenon that Chl d is biosynthesized even in the dark. Under the anoxic dark conditions where Chl d degradation was observed, both growth and new Chl d biosynthesis appear to have stopped. Therefore, Chl d degradation may have been observed because Chl d synthesis had stopped, as you indicated.